# Re-Evaluation of Combinational Efficacy and Synergy of the Italian Protocol In Vitro: Are We Truly Optimizing Benefit or Permitting Unwanted Toxicity?

**DOI:** 10.3390/biomedicines9091190

**Published:** 2021-09-10

**Authors:** Chitra Subramanian, Reid McCallister, Dawn Kuszynski, Mark S. Cohen

**Affiliations:** 1Department of Surgery, Michigan Medicine, Ann Arbor, MI 48109, USA; rmmccall@med.umich.edu (R.M.); dawn.henderlong@yahoo.com (D.K.); 2Department of Pharmacology, University of Michigan, Ann Arbor, MI 48109, USA; 3Department of Biomedical Engineering, University of Michigan, Ann Arbor, MI 48109, USA

**Keywords:** adrenocortical carcinoma, Italian protocol, mitotane, chemotherapeutics

## Abstract

Introduction: Adrenocortical carcinoma (ACC) is a rare endocrine malignancy, with very poor prognosis as a majority of the patients have advanced disease at the time of diagnosis. Currently, adjuvant therapy for most patients consists of either mitotane (M) alone or in combination with multi-drug chemotherapeutics such as etoposide (E), doxorubicin (D), and cisplatin (P), known as the Italian protocol (IP; EDPM). This multi-drug treatment regimen, however, carries significant toxicity potential for patients. One way to improve toxicity profiles with these drugs in combination is to understand where their synergy occurs and over what dosing range so that lower dose regimens could be applied in combination with equal or improved efficacy. We hypothesize that a better understanding of the synergistic effects as well as the regulation of steroidogenic enzymes during combination therapy may provide more optimized combinational options with good potency and lower toxicity profiles. Methods: Two human ACC cell lines, NCI-H295R (hormonally active) and SW13 (hormonally inactive), were grown in 2D culture in appropriate growth medium. The viability of the cells after treatment with varying concentrations of the drugs (E, D, and P) either alone or in combinations with M was determined using the CellTiter Glow assay after 72 h, and the combination index for each was calculated using Compusyn by the Chou–Talalay method. The expression levels of enzymes associated with steroidogenesis were evaluated by RT-PCR in NCI-H295R. Results: When both cell lines were treated with M (ranging 25–50 μM), +E (ranging 18.75–75 μM), and +D (ranging 0.625–2.5 μM) we observed a synergistic effect (CI < 1) with potency equivalent to the full Italian protocol (IP), whereas combining M + P + D had an antagonistic effect (CI > 1) indicating the negative effect of adding cisplatin in the combination. Comparing the hormonally active and inactive cell lines, M + P + E was antagonistic in NCI-H295R and synergistic in SW13. Treatment of NCI-H295R cells with antagonistic combinations (M + P + D, M + P + E) resulted in a significant decrease in the levels of steroidogenic enzymes STAR, CYP11A1, and CYP21A2 compared to IP (*p* < 0.05) while M + E + D resulted in increased expression or no significant effect compared to IP across all genes tested. Conclusions: The synergistic effect for M + E + D was significant and equivalent in potency to the full IP in both cell lines and resulted in a steroidogenic gene expression profile similar to or better than that of full IP, warranting further evaluation. Future in vivo evaluation of the combination of M + E + D (with removal of P from the IP regimen) may lower toxicity while maintaining anticancer efficacy in ACC.

## 1. Introduction

Adrenocortical carcinoma (ACC) is a rare and clinically aggressive endocrine neoplasm of the adrenal cortex with an annual incidence of 0.5–2 cases per million [1,2]. Despite survival improvements in many other malignancies from advances in adjuvant therapy strategies, the prognosis of patients diagnosed with ACC remains dismal, with a 5-year overall survival rate of only 37–47% [3,4,5]. Despite surgical resection of locoregional tumors, greater than 70% of ACC patients relapse and develop metastatic disease [6]. The median survival of patients diagnosed with stage IV metastatic disease is less than a year due to either limited or ineffective treatment options [7].

For over two decades, the adrenolytic agent mitotane, which is a synthetic derivative of the insecticide dichlorodiphenyltrichloroethane (DDT), remains the mainstay of treatment for ACC. Even though the exact anti-cancer mechanism of action of mitotane is not clear, inhibition of 11-β hydroxylation and P450 side-chain cleavage leading to blockage of cortisol synthesis in the mitochondria, and mitochondrial-mediated intracellular stress, are known to play an important role in the effectiveness of mitotane treatment in ACC [8,9]. A recent study, evaluating the efficacy of mitotane as a monotherapy in 127 metastatic, inoperable ACC patients identified median overall survival (OS) and progression-free survival (PFS) as 18.5 months and 4.1 months, respectively, which is significantly better than no treatment [10]. Additional multivariate analysis revealed recurrence of greater than 1 year and low tumor burden of <10 lesions as two favorable factors of mitotane monotherapy as the median OS and PFS for these patients are 29.6 months and 8.8 months, respectively [10].

Despite mitotane being beneficial, advanced inoperable ACC continues to carry a dismal prognosis due to tumor aggressiveness and recurrence after the first-line treatment [11]. In addition to its adrenolytic effects, mitotane is also known to inhibit the multidrug resistance protein MDR-1/P-glycoprotein, thereby enhancing the effectiveness of other chemotherapeutic drugs [12,13]. Given this therapeutic rationale, mitotane in combination with other chemotherapeutic agents including doxorubicin, etoposide, cisplatin, vincristine, 5-fluorouracil, or streptozotocin are commonly used as a second line of treatment for advanced- stage ACC patients. Studies have shown that combination treatment of cisplatin with mitotane [14] resulted in a complete response rate in 30% of ACC, whereas when cisplatin, etoposide, and mitotane were combined, the response rate increased to 33% [15]. In another series of 22 advanced ACC patients, a combination of mitotane with streptozotocin resulted in an overall response rate of 36.4% [16]. To date, the best response rate obtained for any combination therapy involves the Italian protocol, which combines mitotane with etoposide (E), doxorubicin (D), and cisplatin (P), also known as the EDP-M regimen. A large multicenter phase II trial of 72 inoperable ACC patients studying the administration of EDP-M achieved an overall response rate of 48.6% [12]. In the FIRM-ACT trial, EDP-M showed significantly higher anti-tumor efficacy than streptozotocin plus mitotane [17]. Due to this significant response rate of EDP-M compared to any other combination strategies, it is commonly used as a standard treatment regimen for many patients with advanced ACC.

Although EDP-M combination therapy in recent phase III clinical trials was effective in prolonging progression-free survival, it did not produce complete responses or cures, and treatment response was not durable long-term due to either disease progression or toxicities associated with combining these cytotoxic drugs. Several serious side effects of EDP-M include nausea, anorexia, vomiting, diarrhea, lethargy, depression, drowsiness, vertigo, alopecia, heart failure, ataxia, renal toxicity, liver toxicity, ototoxicity, and heart failure [11,18,19]. Hence, there remains a critical unmet need to improve toxicity profiles of combination therapy for ACC while maintaining treatment efficacy. We hypothesize that there are better synergistic combinations of EDP-M that maintain therapeutic efficacy while using lower doses of each drug in combination. Identifying these potential synergies may provide combination doses of these drugs that are more clinically durable due to lower toxicity from lower doses being needed with a goal of improving options for patients with advanced disease who still lack safe, durable treatments.

## 2. Materials and Methods

### 2.1. Cell Lines and Drugs

Two genetically validated ACC cell lines SW13 and NCI-H295R were grown in 2D culture in a humidified atmosphere of 5% CO_2_ in air at 37 °C. SW13 was grown in DMEM medium (Thermo Fisher Scientific, Waltham, MA, USA) supplemented with 10% fetal bovine serum (Thermo Fisher Scientific, Waltham, MA, USA) and 1% antibiotic-antimycotic (Thermo Fisher Scientific, Waltham, MA, USA). NCI-H295R was grown in DMEM/F12 supplemented with 10% fetal bovine serum (Thermo Fisher Scientific, Waltham, MA, USA), 1% antibiotic-antimycotic (Thermo Fisher Scientific, Waltham, MA, USA), and 1% Insulin-Transferrin-Selenium (ITS) (Thermo Fisher Scientific, Waltham, MA, USA).

### 2.2. Viability Assay

To measure luminescent viability, approximately 3000 SW13 cells/well and 5000 NCI-H295R cells/well were plated in a white clear-bottom 96-well plate. Cells were treated with serial dilutions of mitotane(M) from 400 μM, cisplatin (P) from 100 μM, etoposide (E) from 300 μM, and doxorubicin (D) from 5 μM for 72 h. Luminescent viability of the cells post-treatment was measured as per the manufacturer’s instruction (Promega Corporation, Madison, WI, USA) by quantifying the ATP by luminescence after the addition of the CellTiter Glo reagent using a BioTek Synergy Neo plate reader and Gen5 software (BioTek, Winooski, VT, USA). The dose–response curves and the half-maximum inhibitory concentrations (IC_50_) of each drug for both the cell lines were then calculated using GraphPad Prism software (GraphPad Prism Software, Inc., La Jolla, CA, USA).

### 2.3. Combination Treatments

For combination treatment of mitotane with chemotherapeutic agents (E, D, and P), two concentrations of mitotane close to IC_50_ values for NCI-H295R (25 and 50 μM) and SW13 (50 and 75 μM) were used. The drug combinations for chemotherapeutics used were E, D, and P as single agents and P + D, D + E, E + P, and E + D + P along with mitotane. Multiple concentrations of chemotherapeutics based on the IC_50_ values of each compound in SW13 and NCI-H295R cells after 72 h drug treatment were used in the combination studies. Viability was determined after 72 h drug treatment by Cell Titer Glo as above, and the combination index (CI) was determined using the Chou–Talalay [20] equation in CompuSyn Software (Compusyn, Inc., Paramus, NJ, USA). CI values of <1 was considered synergistic and the CI values of >1 were set as antagonistic whereas CI = 1 is additive. Combenefit Software was used to generate surface maps of combination biological effects to evaluate the extent of synergistic/antagonistic effects (Cancer Research UK Cambridge Institute).

### 2.4. Real-Time PCR Analysis of Steroidogenic Enzymes

To evaluate the expression levels of steroidogenic enzymes, RNA was isolated from steroidogenic NCI-H295R cells after treatment with drug combinations for 24 h using the Qiagen RNA isolation mini kit (Qiagen Sciences, Valencia, CA, USA) as per the manufacturer instructions. Approximately 200 ng of RNA was reverse transcribed using the Qiagen one-step RT-PCR kit (Qiagen Sciences, Valencia, CA, USA), and the cDNA was subjected to quantitative polymerase chain reaction (PCR) in a step-1 real-time PCR (RT-PCR) machine using the steroidogenic gene-specific primer sets as previously described [21]. Relative gene expression levels were calculated using the ΔΔCt method after normalization with the internal control GAPDH.

### 2.5. Statistical Analysis

All experiments were repeated twice in triplicate and the values were presented as mean ± standard deviation. The control untreated cells were kept as 100% viable and the viability of the treatment effects was calculated as a percentage of the untreated control. The fold changes in steroidogenic enzymes are calculated in terms of the untreated control. Statistical significance between the mean values was determined by student’s *t*-test and the Bonferroni post hoc test using GraphPad Prism. *p*-values of < 0.05 and < 0.001 are represented as * and ***, respectively.

## 3. Results

### 3.1. Synergistic Effect of Adrenocortical Carcinoma Cells after Treatment with Varying Concentrations of Chemotherapeutic Drugs in the Italian Protocol

To evaluate the effect of synergism of chemotherapeutic drugs used in the Italian protocol, we first evaluated the effect of single drugs, namely mitotane, cisplatin, doxorubicin, and etoposide on NCI-H295R and SW13 cells. The cells grown in 2-D culture were treated with multiple concentrations of chemotherapeutics and then the viability was assessed using Cell TiterGlo after 72 h as per the manufacturer protocol. Our results shown in Figure 1 showed a dose-dependent decrease in viability with increasing concentrations of drugs. Therapeutic ranges of mitotane (M) concentrations derived from these data (50 μM for SW13 and 25 μM for NCI-H295R) were used for the combination studies. We next evaluated the efficacy of each chemotherapeutic compound in combination with mitotane at the therapeutic concentrations at which the cells have greater than 90% viability. When NCI-H295R cells were treated with a combination of mitotane and cisplatin (3.125–25 μM) or etoposide (4.69–37.5 μM) or doxorubicin (0.156–1.25 μM), the percent viability of cells decreased significantly by 12.5–23.6%, 18–22%, and 15–20%, respectively (*p* < 0.001) compared to chemotherapeutic drugs (E or D or P) alone (Figure 2). Whereas, for SW13 cells, the addition of mitotane to cisplatin (1.56 μM–12.5 μM) resulted in a significant reduction in the viability of cells by 30–47% (*p* < 0.01), the addition of mitotane to either E or D had no appreciable change in viability compared to either drug alone (Figure 2). To evaluate whether the combination of mitotane with chemotherapeutics acts synergistically, surface plots were generated in Combenefit (Figure 3). The results from the surface plot as well as the combination index (CI) calculations in Compusyn (Table 1) indicated that combining mitotane with either cisplatin or doxorubicin or etoposide is synergistic for NCI-H295R, whereas only cisplatin plus mitotane was synergistic for SW13 cells.

To evaluate the efficacy of combining two chemo drugs used in the Italian protocol, the viability of ACC cells after treatment with combinations of etoposide + cisplatin (E + P), cisplatin + doxorubicin (P + D), and doxorubicin + etoposide (D + E) in the presence and absence of mitotane was determined using cellTiter Glo (Figure 4A). NCI-H295R and SW13 cells were treated with serial dilutions of drugs (P (25–3.125 μM), D (1.25–0.156 μM), and E (37.5–4.69 μM)) at concentrations higher than their IC50 values plus mitotane at 25 μM and 50 μM, respectively, to determine the efficacy of combination treatment. In SW13 cells, the addition of mitotane to the combinations of E + P and E + D had a significant decrease in cell viability. However, the addition of mitotane to P + D showed no significant change in cell viability at all dosing regimens (*p* < 0.001) (Figure 4A). Treatment of NCI-H295R cells with the combination of P + D and E + P with mitotane resulted in either no change or an increase in viability (*p* < 0.001), while D + E showed decreased cell viability in the presence of mitotane (*p* < 0.005) (Figure 4). When the Italian protocol of E + D + P + M was compared to D + E + M, E + P + M, and D + P + M, our results in both SW13 and NCI-H295R cell lines demonstrated that D + E + M had similar effects as that of the Italian protocol in decreasing viability (Figure 5). Surface plots from the Combenefit analysis of the combination treatments (Figure 6) also showed the best synergy for E + D + M combinations in both ACC cell lines (NCI-H295R and SW13). This indicates that the combination E + D + M may achieve similar efficacy while avoiding some of the side effects of the Italian protocol. Combination indices for all the combinations are given in Table 2.

### 3.2. Effect of Treatment with Varying Concentrations of Chemotherapeutic Drugs in Italian Protocol on Steroidogenesis Enzymes

Mitotane, which is considered the first-line treatment for ACC and is part of the Italian protocol, targets the enzymes involved in steroid hormone synthesis. Additionally, studies have shown downregulation of steroidogenic enzymes CYP11A1, STAR, and CYP17A1 in ACC compared to ACAs [22]. Therefore, we next evaluated changes in steroidogenic genes after treatment of hormone-producing NCI-H295R cells with different combination treatments in the presence of mitotane. Hormone-responsive NCI-H295R cells were treated with chemotherapeutics drugs alone (P or D or E) or in combinations (E + P or D + E or D + P) or the Italian protocol (E + D + P), both in the presence and absence of mitotane. Real-time PCR was used to evaluate expression of the steroidogenic pathway genes CYP17A1, CYP19A1, CYP11A1, CYP11B1, CYP21A2, STAR, and 3β-HSD (Figure 7).

Expression levels of the majority of the steroidogenic enzymes tested were down-regulated when NCI-H295R cells were treated with cisplatin (P) compared to treatment with either doxorubicin (D) or etoposide (E). Compared to mitotane alone, P + M treatment resulted in decreased expression levels of all steroidogenic genes, whereas both D + M and E + M treatment did not alter the majority of genes compared to M. Next, gene expression was tested for combination chemotherapeutics. The addition of cisplatin to both doxorubicin and etoposide resulted in significantly lower lever levels of CYP11A1, CYP21A2, CYP11B1, and STAR, whereas D + P resulted in higher levels of these genes both in the presence and absence of mitotane. When the Italian protocol was compared to treatment with mitotane plus combination therapeutics, D + E + M resulted in no significant change in expression of most genes with significantly increased expression of CYP11B1 and CYP11A1 (*p* < 0.05) while D + P + M and E + P + M resulted in variable effects with significantly decreased expression of multiple genes including STAR, CYP11A1, and CYP21A2 (*p* < 0.05). The above data indicate significant up-regulation of steroidogenic genes, especially CYP11A1 and STAR, for E + D + M. Furthermore, the removal of cisplatin from the EDP-M protocol has a positive effect on steroidogenesis.

## 4. Discussion

Adrenocortical Carcinoma (ACC) is a rare endocrine cancer with a high mortality rate where most patients present with advanced metastatic disease. Despite an increased understanding of the pathophysiology of ACC and standard-of-care treatment protocols in recent years, ACC continues to have a poor prognosis. Genetic and multi-omic studies have identified genes such as insulin-like growth factor 2 (IGF2), Wnt/b-catenin, TP53, and others as potential drivers of ACC leading to the development of targeted therapies for ACC. Despite overexpression of IGF2 in ACC [23], a phase 3 clinical trial with the IGF2 inhibitor linsitinib did not reveal any improvement in overall survival of ACC patients [24]. Similarly, targeted therapies with mTOR inhibitors or tyrosine kinase inhibitors also did not improve overall survival in patients [25,26]. Recent identification of markers of immunotherapy (PD1, PD-L1, microsatellite instability, and mutation burden) in ACC led to clinical trials with immune checkpoint inhibitors (pembrolizumab, nivolumab, avelumab, and ipilimumab) for advanced ACC patients. These trial results were, however, heterogeneous and had relatively low response rates, again with no significant improvement in overall survival [27]. Therefore, with EDP-M as the best therapeutic strategy from an efficacy standpoint, it is important to better understand the synergies and toxicity risks associated with this chemotherapeutics. As such, we investigated variations of these combinations over multiple dose levels to define which combinations may be additive, antagonistic, or synergistic in ACC cell lines in vitro. Our results indicate that combining cisplatin with mitotane and etoposide has an antagonistic effect in combination while combining doxorubicin with mitotane and etoposide has a synergistic effect (removing the cisplatin from the combination). Compared with 4-drug EDP-M, which is the standard IP regimen, only the combination of mitotane, etoposide, and doxorubicin showed comparable or improved effects on cell viability in both cell lines tested. The effects of mitotane alone versus mitotane in combination with cisplatin and etoposide are being investigated within an ongoing clinical trial for surgical ACC patients with high risk of recurrence—again speaking to the need for better characterization of combination chemotherapeutic options for ACC (24). Additionally, our findings offer a rationale that can be tested translationally in future in vivo studies, and if these show a similar benefit, this can potentially lower some of the toxicity associated with including cisplatin in the combination and provide a guide for further clinical combinational strategies for patients ineligible to receive standard-of-care treatment [28]. If new combinations with less cytotoxic drugs or doses can be utilized with equal or better efficacy, this could provide more options for all patients with aggressive ACC.

Mitotane treatment results in a significant decrease in adrenal steroidogenesis requiring enzyme replacement therapy [29]. Our in vitro data indicate that combination therapy with mitotane, doxorubicin, and etoposide (E + D + M) had higher levels of steroidogenic precursor genes while combinations that included cisplatin (E + P + M or D + P + M) failed to achieve similar results. These data indicated that treatment with E + D + M had a much lower toxic effect on adrenal steroidogenesis than any combination including cisplatin. In thinking about the rationale for this effect, one explanation comes from the differential effect these drugs have on the steroid synthesis pathway. The reduced STAR, CYP11A1, and CYP21A2 observed with either MPD or MPE treatment may be due to the effect that cisplatin has on the steroid synthesis pathway—in essence, down-regulating these genes. In contrast, doxorubicin upregulates these steroid synthesis genes. This antagonistic effect between doxorubicin and cisplatin on steroid synthesis would also explain in part why MED treatment is able to increase the enzymatic expression over EDP-M since MED does not have the down-regulating effect from cisplatin present compared to EDP-M. Additional steroidal pathway investigation in future studies would better validate this mechanism.

Each of these important in vitro findings indicate that combination therapy with mitotane, etoposide, and doxorubicin (E + D + M) may achieve similar efficacy to the standard Italian Protocol (E + D + P + M), while avoiding the toxicities generated from adding cisplatin to the treatment combination. It is important to also note that combining cisplatin with doxorubicin is agonistic in tumors like SW13, whereas cisplatin either alone or in combination with etoposide or doxorubicin was antagonistic in tumors with secreting capacity like H295R. This suggests a differential effect based on a tumor’s secreting capability, which will need additional in vivo validation, and further translational evaluation of the E + D + M regimen is required due to the limitations of this study being in vitro and on homogeneous immortalized cell lines. It will be important in future studies to translate and validate these findings in patient-derived xenografts that contain more cellular heterogeneity typical of aggressive ACCs. Further translational evaluation of the E + D + M regimen is required due to the limitations of this study being in vitro and on homogeneous immortalized cell lines. It will be important in future studies to translate and validate these findings in vivo and in patient-derived xenografts that contain more cellular heterogeneity typical of aggressive ACCs. Additionally, we will need to evaluate additional dose combination levels to optimize which dosing combinations will have the highest synergy in vivo. Our findings, however, are promising and represent a novel evaluation of the synergistic effects of combination therapy in ACC. Further evaluation and translation of these findings are needed to provide preclinical support for novel combination therapies clinically that decrease toxicity, increase durability, and improve upon the efficacy of the Italian protocol.

## Figures and Tables

**Figure 1 biomedicines-09-01190-f001:**
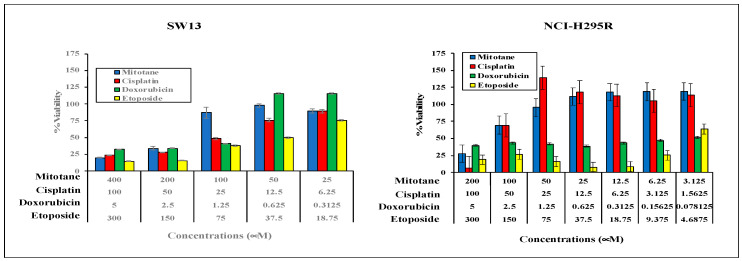
Two ACC cell lines (NCI-H295R and SW13) were treated with varying concentrations of either mitotane, cisplatin, or doxorubicin or etoposide for 72 h and the viability of cells was measured by MTS assay. Percentage viability of cells with respect to untreated control was calculated. Experiments were repeated twice in triplicate, and the values are presented as mean ± standard deviation.

**Figure 2 biomedicines-09-01190-f002:**
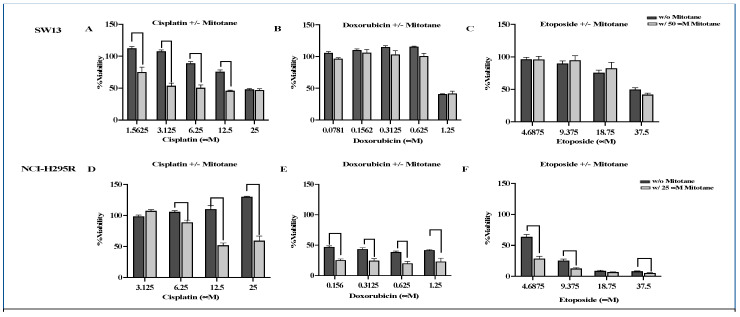
SW13 and NCI-H295R cells were treated with chemotherapeutic drugs (etoposide (**C**,**F**), doxorubicin (**B**,**E**), or cisplatin (**A**,**D**)) with or without mitotane and the viability was determined by MTS assay. Percentage viability of cells with respect to untreated control was calculated. Experiments were repeated twice in triplicate, and the values are presented as mean ± standard deviation.

**Figure 3 biomedicines-09-01190-f003:**
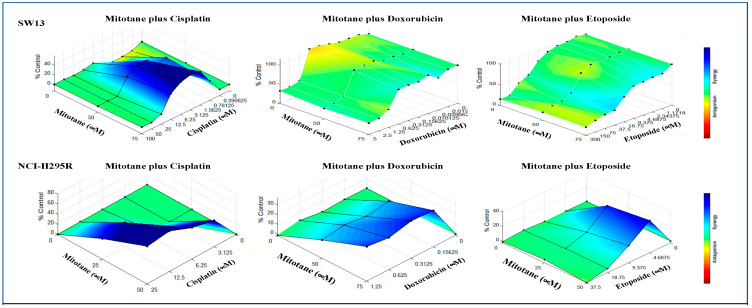
Combenefit analysis showing synergistic combinations of mitotane and one other chemotherapeutic drug (M + P, M + E. and M + D). Surface maps of combination effects plotted using Combenefit software illustrating color coded maps for synergistic/antagonistic effects.

**Figure 4 biomedicines-09-01190-f004:**
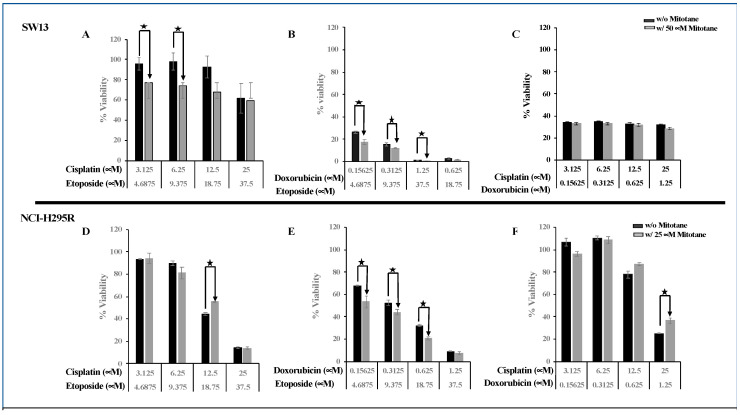
SW13 and NCI-H295R cells were treated with two chemotherapeutic drugs (P + E (**A**,**D**) or P + D (**C**,**F**) or E + D (**B**,**E**)) with or without mitotane used in the Italian protocol and the viability was determined by MTS assay. Percentage viability of cells with respect to untreated control was calculated. Experiments were repeated twice in triplicate, and the values are presented as mean ± standard deviation. * *p* < 0.05.

**Figure 5 biomedicines-09-01190-f005:**
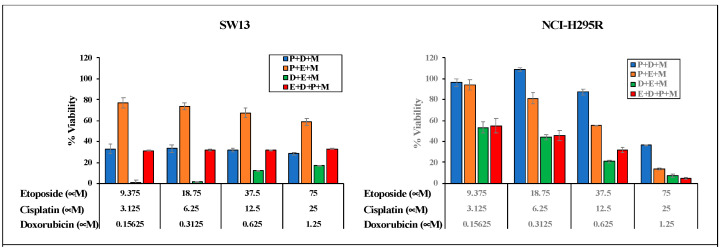
Comparison of viability of chemotherapeutic drug combinations (M + P + E or M + P + D or M + E + D) and Italian protocol (E + D + P + M) in NCI-H295R and SW13 cells.

**Figure 6 biomedicines-09-01190-f006:**
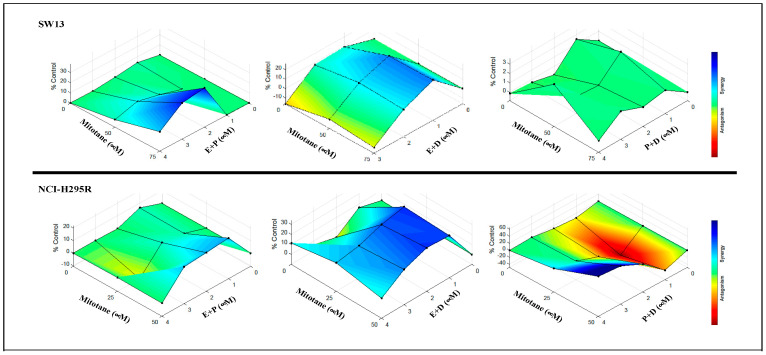
Combenefit analysis showing synergistic combinations of mitotane with combinations of chemotherapeutics (M + P + E or M + P + D or M + E + D) in Italian protocol. Surface maps of combination effects plotted using Combenefit software illustrating color coded maps for synergistic/antagonistic effects.

**Figure 7 biomedicines-09-01190-f007:**
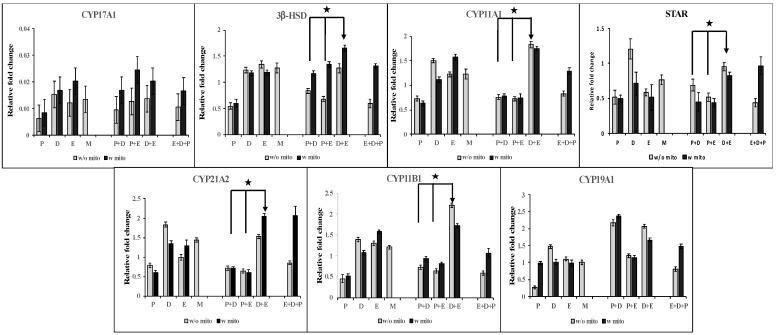
Expression levels of several enzymes involved in steroidogenesis pathway. Hormonally active NCI-H295R cells were treated with concentrations of Chemotherapeutics as for 24 h and expression levels of enzymes was measured by RT-PCR. *p* < 0.05 is represented as *.

**Table 1 biomedicines-09-01190-t001:** ACC cell lines NCI-H295R (Left) and SW13 (Right) were treated with varying concentrations of chemotherapeutics with or without mitotane for 72 h and the combination Index was calculated using Chou-Talalay equation. C < 1 is synergistic, C = 1 additive, and C > 1 is antagonistic (Red text).

NCI-H295R			SW13		
**Mitotane**	**Cisplatin**	**CI**	**Mitotane**	**Cisplatin**	**CI**
25	25	0.49291	50	25	0.49291
	12.5	0.25187		12.5	0.25187
	6.25	0.72284		6.25	0.72284
	3.125	3.95163		3.125	3.95163
50	25	0.22882	75	25	0.22882
	12.5	0.12536		12.5	0.12536
	6.25	0.1907		6.25	0.1907
	3.125	0.22456		3.125	0.22456
**Mitotane**	**Doxorubicin**	**CI**	**Mitotane**	**Doxorubicin**	**CI**
25	1.25	0.0428	50	1.25	0.46056
	0.625	0.03738		0.625	14.7115
	0.3125	0.04454		0.3125	13.1713
	0.15625	0.04637		0.15625	12.4012
50	1.25	0.02699	75	1.25	0.65588
	0.625	0.03741		0.625	3.0185
	0.3125	0.02337		0.3125	3.56345
	0.15625	0.04418		0.15625	3.17433
**Mitotane**	**Etoposide**	**CI**	**Mitotane**	**Etoposide**	**CI**
25	37.5	0.01439	50	37.5	0.69421
	18.75	0.01629		18.75	2.37633
	9.375	0.04815		9.375	6.14957
	4.6875	0.68287		4.6875	8.35123
50	37.5	0.0665	75	37.5	0.6226
	18.75	0.02978		18.75	0.96852
	9.375	0.01581		9.375	1.81114
	4.6875	0.2288		4.6875	1.74636

**Table 2 biomedicines-09-01190-t002:** (**A**,**B**) ACC cell lines NCI-H295R (Left) and SW13 (Right) were treated with varying concentrations of combinations of chemotherapeutics without ((**A**)-Top) or with ((**B**)-Bottom) mitotane for 72 h and the combination Index was calculated using Chou-Talalay equation. C < 1 is synergistic, C = 1 additive, and C > 1 is antagonistic (Red text).

**(A)**							
**Drug Concentrations (M)**				**Combination Index (CI)**			
**Doxorubicin**		**Etoposide**		**SW13**		**NCI-H295R**	
1.25		37.5		0.05704		0.07368	
0.625		18.75		0.04269		0.52351	
0.3125		9.375		0.07947		1.10422	
0.15625		4.6875		0.06444		0.9597	
**Cisplatin**		**Etoposide**		**SW13**		**NCI-H295R**	
25.0		37.5		2.14524		0.30695	
12.5		18.75		4.76774		1.38877	
6.25		9.375		6.28293		36.0221	
3.125		4.6875		1.8226		11.182	
**Cisplatin**		**Doxorubicin**		**SW13**		**NCI-H295R**	
25.0		1.25		4.70672		0.86499	
12.5		0.625		2.41979		4.23615	
6.25		0.3125		1.26733		626.994	
3.125		0.15625		0.62515		313.497	
**(B)**							
**SW13**							
**Mitotane**	**Doxorubicin**	**Cisplatin**	**CI**	**Mitotane**	**Doxorubicin**	**Cisplatin**	**CI**
50.0	1.25	25.0	0.75264	75.0	1.25	25.0	0.89046
	0.625	12.5	0.47816		0.625	12.5	0.51558
	0.3125	6.25	0.31046		0.3125	6.25	0.36737
	0.15625	3.125	0.21563		0.15625	3.125	0.29093
**Mitotane**	**Doxorubicin**	**Etoposide**	**CI**	**Mitotane**	**Doxorubicin**	**Etoposide**	**CI**
50.0	1.25	37.5	0.04192	75.0	1.25	37.5	0.04498
	0.625	18.75	0.03672		0.625	18.75	0.03092
	0.3125	9.375	0.1072		0.3125	9.375	0.09039
	0.15625	4.6875	0.10291		0.15625	4.6875	0.09362
**Mitotane**	**Cisplatin**	**Etoposide**	**CI**	**Mitotane**	**Cisplatin**	**Etoposide**	**CI**
50.0	25.0	37.5	2.00493	75.0	25.0	37.5	1.50331
	12.5	18.75	2.6954		12.5	18.75	1.36302
	6.25	9.375	3.86777		6.25	9.375	1.03401
	3.125	4.6875	11.5887		3.125	4.6875	88.4208
**NCI-H295R**							
**Mitotane**	**Doxorubicin**	**Cisplatin**	**CI**	**Mitotane**	**Doxorubicin**	**Cisplatin**	**CI**
25.0	1.25	25.0	1.71632	50.0	1.25	25.0	1.33159
	0.625	12.5	11.8672		0.625	12.5	16.9827
	0.3125	6.25	634.151		0.3125	6.25	6.4654
	0.15625	3.125	23.7892		0.15625	3.125	3.10297
**Mitotane**	**Doxorubicin**	**Etoposide**	**CI**	**Mitotane**	**Doxorubicin**	**Etoposide**	**CI**
25.0	1.25	37.5	0.07845	50.0	1.25	37.5	0.06546
	0.625	18.75	0.24998		0.625	18.75	0.25894
	0.3125	9.375	0.74028		0.3125	9.375	0.33202
	0.15625	4.6875	0.73815		0.15625	4.6875	0.50166
**Mitotane**	**Cisplatin**	**Etoposide**	**CI**	**Mitotane**	**Cisplatin**	**Etoposide**	**CI**
25.0	25.0	37.5	0.34045	50.0	25.0	37.5	0.40241
	12.5	18.75	2.87857		12.5	18.75	0.054566
	6.25	9.375	10.9307		6.25	9.375	4.00527
	3.125	4.6875	14.6393		3.125	4.6875	4.39413

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
