# Peer review of "Re-Evaluation of Combinational Efficacy and Synergy of the Italian Protocol In Vitro: Are We Truly Optimizing Benefit or Permitting Unwanted Toxicity?"

_biomedicines, 2021, doi:10.3390/biomedicines9091190_

Round 1

Reviewer 1 Report

The Article “Re-evaluation of combinational efficacy and synergy of the
Italian Protocol in vitro: Are we truly optimizing benefit or permitting unwanted toxicity?  By Chitra Subramanian et al  is a very interesting article that will attract the attention of readers interested in adrenocortical carcinomas biology as well as in ACC medical treatment.

It is an additional evidence that using more drugs doesn’t necessarily mean getting better results!

The first question that I would like the authors to comment on, is: After this study, do they feel that there is strong evidence to remove cisplatin from the EDP-M protocol? (this would have better results with lesser toxicity…)

The authors evaluated the synergism/antagonism of chemotherapeutic drugs used in what they called the Italian Protocol comparing the effect of single drugs and then of combinations by assessing cell viability. Why did they not assess cell proliferation or apoptosis?

Could this study conclusions mean that Cisplatin is antagonizing the other treatments when the tumors have secreting capacity (as in H295R cell line) but it can be positive in non-secreting less differentiated tumors like in SW13 cell line?

Is there any Randomized Clinical Trial going on, or programmed, comparing ED-M with EDP-M? The RCT reported in Ref 24 compares M with MPE but apparently a comparison of M with MED would be more warranted.

Concerning the steroidogenic enzymes that were studied:

MPD and MPE significantly reduced StAR, CYP11A1 and CYP21A2 compared to EDP-M. Can the authors attempt to provide any explanation for this? And in parallel why does MED increase the enzymatic expression and what can be the consequences of that?

The article is well written in spite of a few English incorrections like for instance in lines 323-325:

“Since mitotane, which is considered first-line treatment for ACC by many, and also is part of the Italian Protocol, is known to target the enzymes involved in the steroid hormone synthesis.” (this sentence as is makes no sense. Either the authors remove the word “Since” or re-phrase it more extensively)

Or like in Reference 1 where you find an author named Medonca which is in fact Mendonça.

And speaking of References a few references are missing and the paper might benefit from including them:

Fassnacht M et al published the results of a RCT (FIRM-ACT) comparing EDP-M and streptozotocin + Mitotane, in 2021 (NEJM). This landmark paper is very much related to the present article

S.Pereira et al published a paper on the biology of IGF2 in ACC in Endocrine (doi 10.1007/s12020-019.02033 5) a study that was performed in H295R cell line, as the present article, in which the authors demonstrate that IGF2 increases cell viability as well as proliferation and inhibition of mTOR reverted IGF2 effects on cell viability and proliferation.

Those same authors recently published another article, in Biomedicines 2020, named “Incomplete Pattern of Steroidogenic protein expression in functioning Adrenocortical carcinomas”(doi: 103390/biomedicines 8080 256) which is highly relevant to the discussion of the present article that presents results of the effects of chemotherapeutic drugs precisely in these enzymes.

Author Response

We ave made requested changes to the manuscript as per the reviewer comments.

Reviewer 2 Report

In this study the authors tested the best combination treatment for adrenocortical carcinoma (ACC) using two human ACC cell lines. They have shown that the combination mitotane+ etoposide + doxorubicine has the same efficacy (with less toxicity) of the full Italian protocol in both ACC cell lines, suggesting the removal of cisplatin from the protocol. The methods are well described and the results are clear. However, the quality of figures can be improved:

  • in Figure 1 use the same y-axis range for both panels
  • in Figure 2 use the same y-axis range for panels D, E, F
  • in Figure 4 use the same y-axis range for all panels.

Please comment in the discussion section similar data of other research groups if available.

Author Response

Thank you. The figures were modified as per the reviewer comments.
